# Recurrent Neural Networks as Dynamical Systems: Exploring Chaos and Order

**Pravish Sainath**

20125633

Département d'informatique et de recherche opérationnelle (DIRO)

Université de Montréal

Montréal, QC

`pravish.sainath@umontreal.ca`

## Abstract

Chaos is a property of many dynamical systems. The impact of this phenomenon of chaos on recurrent neural networks (RNNs) is studied based on prior work. Some problems with the most common RNN architectures due to chaos and instability are analyzed theoretically and empirically. These are related to having strange attractors and the consequent vanishing/exploding gradient problems that make it difficult to learn long-term dependencies. Specialized architectures from the literature such as Chaos-Free RNN (CF-RNN) and Antisymmetric RNN (AS-RNN) claim to help mitigate these issues and ensure that RNNs learn good attractors. A novel variant of CF-RNN, called the Asymmetric Chaos-Free RNN (AS CF-RNN) is proposed and studied. Some empirical analysis is carried out to examine the advantages brought by each of these architectures. This is a learning attempt to view RNNs through the lens of dynamical systems and understand them in perspective.

## 1 Introduction

It has been known since many years that artificial neural networks (ANNs) exhibit dynamics. Recurrent Neural Networks (RNNs), especially are a class of universal function approximation models [20], that have been widely used in several applications related to sequential modeling. They have been viewed as dynamical systems since their conception [3, 12]. Their success can be attributed to the fact that they approximate the temporal dynamics of systems over long periods of time.

In recent years, many dynamical systems tools have started being utilized in the study of these ANNs. Especially, understanding the aspects of chaos and stability (*order*) of dynamical systems are of particular interest to discover good models [4]. *Chaos* is referred to the sensitivity to initial conditions of many dynamical systems that leads to aperiodic orbits due to their natural characteristic. Over several years, it has been known that ANNs (including random networks) exhibit this phenomenon [21], that can potentially make learning difficult for them in some cases. Stability issues in RNN training such as exploding or vanishing gradients manifest themselves in the models, causing problems in consolidating inputs in the long-term [3, 19].

This report summarizes the notion of chaos in RNNs and explores some ways to avoid it in many contexts. As a part of this study, firstly, the characteristics of RNNs that exhibit and do not exhibit chaos have been reproduced and compared. Secondly, a derivation of a stable hybrid architecture (based on existing ones from literature) through an analytical treatment has been attempted. Finally, the new architecture is trained for a task and its performance is compared with the other models to understand the impact of stability on learning long-range dependencies.

Final Project for the course: MAT 6215 - Dynamical Systems (Winter 2021)

## 2 Background and Related Work

The work done for this study is based on the ideas drawn from *Laurent et al* [15] and *Chen et al* [7].

RNNs can be viewed as a discrete [13] or a continuous [16, 14] dynamical systems with their respective model parameters. Formulations exist for both of these cases and they have been studied for various applications. While analyzing RNNs using the dynamical systems framework, it is very common to consider that the network does not receive any input and only the hidden state dynamics happen [15]. This is called the dynamical system *induced by the RNN* This setup views inputs as perturbations to the system that is trying to achieve certain dynamics. This assumption also makes the network an autonomous system (with time-dependent inputs not considered). Also, it helps to decouple the effects of the input and the state.

The basic theory about RNNs as dynamical systems is that they learn good attractors in order to process the patterns in the input and hence *remember* them [3]. We know that dynamical systems that are non-linear and/or recursive in nature have a possibility of exhibiting chaotic behavior for certain settings of the system parameters (example, a logistic map with the parameter $k > 3.8$). Similarly, as RNNs have model parameters represented by their weight matrices, that determine the parameters of the function iteratively computed by them. For certain values of these weights, they may exhibit chaotic behavior [15]. Strange attractors that are generated due to chaos, lead to the exponential divergence of trajectories of any two distinct but neighboring states. Thus the network cannot learn predictable associations. Many weight initialization schemes aim to avoid this chaotic regime. In [15], they show that the dynamical system induced by an LSTM or GRU can exhibit chaotic behavior and they propose a new architecture called the **Chaos-Free RNN** (CF-RNN) to stabilize the induced dynamics based on an impulse response model with two gates - input gate $\theta$ and forget gate $\eta$.

The most useful way of bringing more order in RNNs is using stability analysis of dynamical systems to understand and control their behavior in the long-term [18]. Unstable systems cause the gradients to explode, while stable sustems can still experience vanishing gradients, resulting in forgetting of long-rage input [12]. Some studies use gated networks [12, 8], while others such as [11, 17] have utilized constraints on the parameter matrix of RNNs to address this. With these two being the most common approaches, there are other methods that deal with this using a norm-based regularization [22] or skip-connections [10] or preventing non-linearity saturation [5]. **Antisymmetric RNN** [6] (AS-RNN) ensure the stability by constraining the parametric weight matrix to be antisymmetric. This property bounds the gradients and ensures memory of long-range inputs [13, 18].

## 3 Proposed Architecture - Derivation

In this section, a variant of the CF-RNN [15] architecture is reverse-engineered by applying the principles of stability used in AS-RNN [6]. The gated version of AS-RNN stated in [6], despite being stable, only has a single gate that controls both the input and the state. On the other hand, CF-RNN has distinct gates for the input and state. The purpose of this architecture is to study the impact of stability on its long-term learning functions. The *local stability* property of the AS-RNN imposed by the asymmetric weight matrix can be utlized to deduce the equations of *a* dual-gated architecture based on CF-RNN, to improve it further.

Eq. 1 lists the expressions for the forget gate $\theta$, the input gate $\eta$ and the hidden state $h$ at each time step $t$ in terms of the respective weight parameters for a typical CF-RNN.

$$\theta_t = \sigma(W_\theta h_{t-1} + U_\theta x_t + b_\theta) \tag{1}$$
$$\eta_t = \sigma(W_\eta h_{t-1} + U_\eta x_t + b_\eta) \tag{2}$$
$$h_t = \theta_t \odot tanh(h_{t-1}) + \eta_t \odot tanh(U_h x_t) \tag{3}$$

The discrete-time formulation of the dynamics of the state $h$ in eq. 3 can instead be expressed as a continuous-time equation by replacing the $h_t$ by the time derivative $h'_t$. With an assumed initial state $h_0$, this can be expressed as the ODE given in eq. 4.

$$h'_t = \theta_t \odot tanh(h_{t-1}) + \eta_t \odot tanh(U_h x_t), h(0) = h_0 \tag{4}$$

In eq. 3, the function $f$ that corresponds to the ODE system can be expressed as :

$$f(h_{t-1}; x_t, W_h, U_\theta, U_\eta, U_h) = h'_t = \theta_t \odot tanh(h_{t-1}) + \eta_t \odot tanh(Ux_t) \quad (5)$$

The stability of the solutions of the system in eq. 4 is given determined by the eigen values of the Jacobian of the function $f$, that is given in eq. 6. The computation of the Jacobian $J(t) = \frac{\partial f}{\partial h_{t-1}}$ is done in Section B.

$$J(t) = diag\Big(\theta_t \odot (1 - \theta_t) \odot tanh(h_{t-1})\Big).W_\theta + diag\Big(tanh'(h_{t-1}) \odot \theta_t\Big)$$
$$+diag\Big(\theta_t \odot (1 - \theta_t) \odot tanh(U_h x_t)\Big).W_\eta \quad (6)$$

As seen in eq. 6, the Jacobian is a sum of diagonal matrices are multiplied with different matrices $W_\theta, I$ and $W_\eta$ respectively. The nature of eigen values of this Jacobian can give rise to different effects in the stability of the system.

A new expression $J^*(t)$ can be constructed by setting $W_\theta = W_\eta = P$ and post-multiplying the second diagonal matrix in the sum by $P$ as shown in eq. 7:

$$J^*(t) = diag\Big(\theta_t \odot (1 - \theta_t) \odot tanh(h_{t-1})\Big).P + diag\Big(tanh'(h_{t-1}) \odot \theta_t\Big).P$$
$$+diag\Big(\theta_t \odot (1 - \theta_t) \odot tanh(U_h x_t)\Big).P \quad (7)$$

$$\implies J^*(t) = diag\Big(\theta_t \odot (1 - \theta_t) \odot tanh(h_{t-1}) + tanh'(h_{t-1}) \odot \theta_t + \theta_t \odot (1 - \theta_t) \odot tanh(U_h x_t)\Big).P \quad (8)$$

It is known that the eigen values of the product of an invertible diagonal matrix and an antisymmetric matrix are purely imaginary (Proof in Proposition 3 of [6]). The diagonal matrix can be assumed to be invertible (numerically). Thus, this type of product can be achieved in eq. 8 by setting $P = W_h - W_h^T$ for some matrix $W_h$ ($P$ is always antimsymmetric). Thus, the eigen values of the constructed Jacobian $J^*(t)$ are imaginary, giving $Re(\lambda_i(J(t))) = 0$.

The originally assumed model in eq.4 computes the derivative $h'(t)$. For computing the next state $h(t)$, numerical methods used to solve ODEs need to be used. Following the approach of [6], the forward Euler method can be applied for this. However, this method is numerically stable only when the eigen values $\lambda_i$ and the step size $\epsilon$ satisfy the condition $\max\limits_{i=1,2,...,n} |1 + \epsilon\lambda_i(\mathbf{J}(t))| \leq 1$ (Section. C.5).

This can be ensured by using the diffusion constant $\gamma \in [0,1]$ to subract the diagonals of $P$ as given in [6]. The matrix $P$ can thus be modified as $P = (W_h - W_h^T - \gamma I)$. Also, to account for the $P$ that was multiplied in the second term in eq.7, the $tanh(h_{t-1})$ in eq. 3 should become $tanh(Ph_{t-1})$. This ensures that $J^*(t)$ is the Jacobian of the modified function $f^*$ (from $f$ in eq. 5) given by eq. 9:

$$f^*(h_{t-1}; x_t, W_h, U_\theta, U_\eta, U) = h'(t) = \theta_t \odot tanh((W_h - W_h^T - \gamma I)h_{t-1}) + \eta_t \odot tanh(Ux_t) \quad (9)$$

The final equations of the proposed dual-gated Antisymmetric Chaos-Free RNN (AS CF-RNN) obtained after applying a forward Euler step of the new ODE, with a step size $\epsilon$ are given in eq. 10 :

$$\theta_t = \sigma((W_h - W_h^T - \gamma I)h_{t-1} + U_\theta x_t + b_\theta)$$
$$\eta_t = \sigma((W_h - W_h^T - \gamma I)h_{t-1} + U_\eta x_t + b_\eta)$$
$$h_t = h_{t-1} + \epsilon\,\theta_t \odot tanh((W_h - W_h^T - \gamma I)h_{t-1}) + \epsilon\,\eta_t \odot tanh(Ux_t) \quad (10)$$

# 4    Experiments and Results

Two sets of experiments - one for visualizing chaos in RNNs and what it means to be free from them (in Section. 4.1) and the other for empirically verifying the performance of the theoretically stable model AS CF-RNN (in Section. 4.2).

## 4.1    Analysis of chaos : Attractors and Bifurcations

In this experiment (based on [15]), different RNNs with 2-dimensional hidden state vectors were considered and some specific weight matrices (Table. A.1 for the equations and Section. D.1 for the values). These models were first used in the absence of any input (zero input) to generate trajectories from many initial states and plot the attractor on the phase plane, based on [15].

It can be remarked from Fig. 1(a),(b) and (c) that the chosen LSTM and GRU learn strange (chaotic) attractors while the CF-RNN learned only a point attractor at 0.

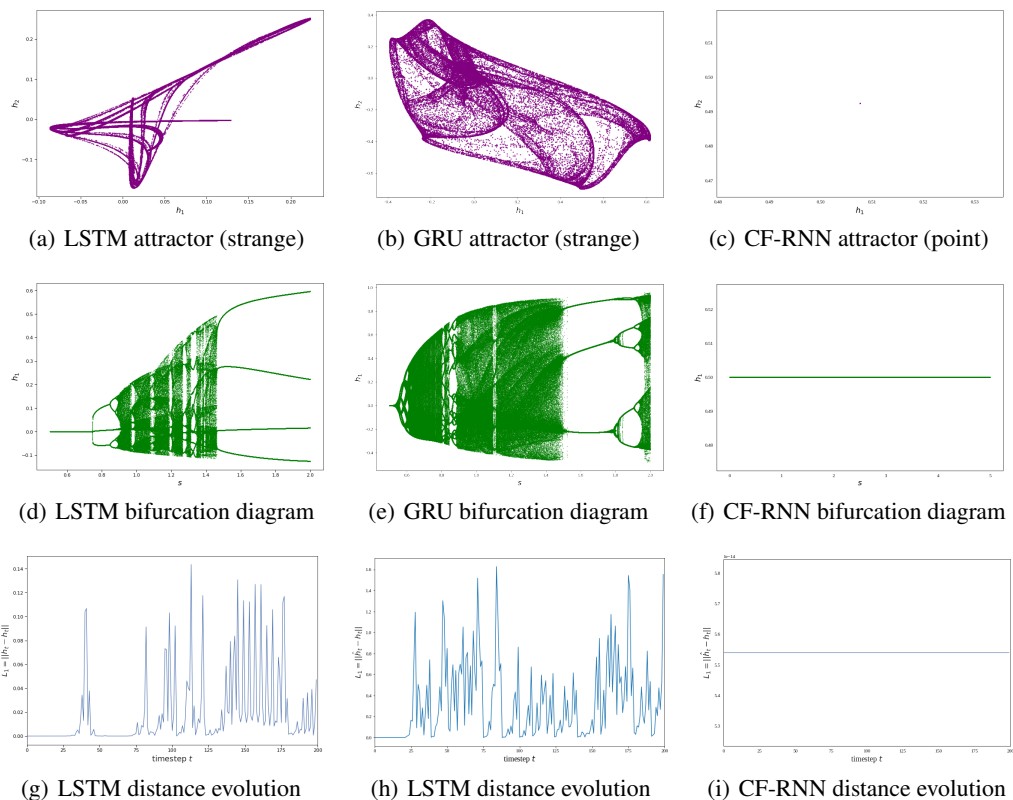

(a) LSTM attractor (strange)    (b) GRU attractor (strange)    (c) CF-RNN attractor (point)

(d) LSTM bifurcation diagram    (e) GRU bifurcation diagram    (f) CF-RNN bifurcation diagram

(g) LSTM distance evolution    (h) LSTM distance evolution    (i) CF-RNN distance evolution

Figure 1: Characteristics of the dynamical systems of LSTM, GRU and CF-RNN chosen for the experiments: (g),(h),(i) - distance plot of $||h_t - \hat{h}_t||^2$ for proximate initial points $h_0$ and $\hat{h}_0$

Using a simple scaling parameter $s$ of the hidden-hidden weight matrix of each model, a bifurcation diagram is created using the states generated by varying the value of the scale $s$ in the system $f(s.W_h)$. The LSTM and GRU show period doubling and eventually chaos for a good range of the parameter. Thus, even a simple scaled version of the weight matrix can produce chaos in the absence of inputs for LSTM and GRU, while CF-RNN has no bifurcations and is consistently simple.

When two states start from very close initial points $h_0$ and $\hat{h}_0$, Fig. 1(g),(h) and (i) show the time evolution of the distance (in phase space) between the states $h_t$ and $\hat{h}_t$ in the respective trajectories. Clearly, from the plot, they move away from each other for chaotic systems (LSTM,GRU here) and remain at fixed distance for CF-RNN. Thus, the chaotic phenomena of dynamical systems induced by LSTMs and GRUs are clearly seen and the CF-RNN has no chaos at all.

## 4.2 Analysis of Proposed Model AS CF-RNN : Sequential tasks

The AS CF-RNN model has been designed based on theoretical stability guarantees. It is interesting to see how much it is able to fulfill this in practice and how well it learns long-term dependencies. This is verified training the network on standard experiments used for testing long-term task, such as the following.

**Copy Task**    Among a set of $K$ symbols, those that need to be copied are presented in the first $N$ time steps, then must be output one by one after a long delay of $T$ time steps (introduced in [12] and explained in [1]). We use $N = 25$ and $K = 19$. The network is trained using the cross-entropy error on the values it predicts.

Table. 1 shows the performance for different values of $T$ of each model. Clearly, the CE of all models is either increasing or constant as $T$ increases as inputs need to be held for longer-ranges. AS CF-RNN and AS-RNN (Gated) features among the best models.

Table 1: Results: The Cross Entropy error (CE) and Accuracy % in the Copy Task of various models for different settings of $T$

| Architecture | | $T = 100$ | | $T = 200$ | | $T = 400$ | | $T = 800$ | |
|---|---|---|---|---|---|---|---|---|---|
| Model | $\gamma$ | CE | Acc % | CE | Acc % | CE | Acc % | CE | Acc % |
| AS CF-RNN | 0.005 | 0.001 | 100 | 0.001 | 100 | 0.001 | 100 | 0.003 | 98.9 |
| AS CF-RNN | 0.1 | 0.002 | 100 | 0.001 | 100 | 0.002 | 100 | 0.005 | 97.2 |
| AS-RNN (Gated) | 0.005 | 0.000 | 100 | 0.001 | 100 | 0.002 | 100 | 0.004 | 98.1 |
| AS-RNN (Gated) | 0.1 | 0.002 | 100 | 0.003 | 98.7 | 0.004 | 97.8 | 0.008 | 94.3 |
| CF-RNN | - | 0.004 | 98.4 | 0.004 | 98.2 | 0.018 | 87.5 | 0.014 | 70.7 |
| LSTM | - | 0.005 | 98.0 | 0.076 | 68.2 | 0.098 | 45.7 | 0.116 | 36.6 |
| GRU | - | 0.006 | 97.8 | 0.085 | 50.2 | 0.125 | 31.1 | 0.535 | 20.2 |

The experiment is repeated for different values of $T$ and the eigenvalues of the end-to-end Jacobian matrix of the RNN $\frac{\partial h_T}{\partial h_0}$ are collected. The absolute values of all the eigen values are used to compute the mean and standard deviation as shown in Figure. 2 (similar to [6]).

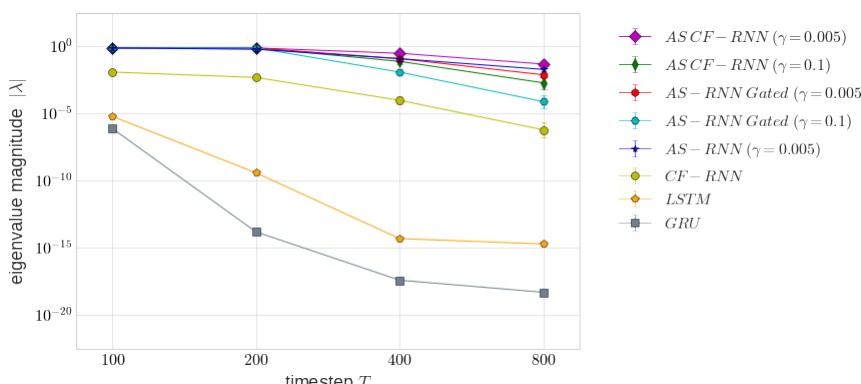

Figure 2: The distribution of eigen values (mean and standard deviation) of the end-to-end Jacobian trained on the Copy Task for different values of $T$ as indicated.

As seen in Figure. 2, the mean eigenvalue magnitude is closer to one for more stable models such as AS CF-RNN (Gated) and AS CF-RNN. For LSTM and GRU models, as $T$ increases, the magnitude shifts lower and lower than 1, tending towards zero, indicating vanishing gradients. The value also reduces for CF-RNN, whereas for its asymmetric counterpart AS CF-RNN, it is almost consistently at the same value. The decay in this value for AS CF-RNN is slower compared to the original CF-RNN model. This indicates that vanishing gradients occur in CF-RNN earlier in time than AS CF-RNN.

Thus, from both Table. 1 and Figure. 2, it can be inferred about the stronger ability of AS CF-RNN to learn long-term dependencies over CF-RNN. AS CF-RNN seems to be fairly comparable to the Gated AS-RNN models with lower $\gamma$.

# 5 Conclusion and Future Work

By viewing RNNs within the framework of dynamical systems, many tools are made available to understand and improve these networks. For example, we understand the space learned by te and analyzing the attractors and bifurcations of some simple networks, we see the necessity of the training to not get into the chaotic regime during learning. There is a clear need for models to learn simple attractors. There are no unpredictable dynamics in such models, making them sensitive to only the impact of the patterns in the data.

The chaos-free network (CF-RNN) leads to simple attractor states and eliminates any scope for chaos. We saw that operating in a chaos-free setting does not necessarily guarantee good learning by the network. Additional stability considerations need to be taken into account to ensure that the gradients are controlled. For example, the aspect of stability in RNNs is key in identifying properties of the dynamics and fixing issues such as exploding and vanishing gradients.

Asymmetric Chaos-Free RNN (AS CF-RNN) was formulated and proposed based on the stability theory. Elements of stability from the dynamical systems theory were applied to an existing model (CF-RNN) with the view of enforcing a certain *order* as a prior in the network to improve its ability to learn long-term dependencies, validated by by both theory and experiments. We see that it is able to effectively gate information from the input as well as the past state, while learning to remember inputs over a longer time range.

The usage of the AS CF-RNN here was mostly for the purpose of understanding the methods and mechanisms of some RNNs. More experiments and analyses are necessary in the future for further study regarding its properties and uses. Some extended analysis of the use of the anti-symmetric. Thus, an existing version of RNN was adapted using dynamical systems formulation in order to obtain a potentially superior model that seems to work at least fairly as well for learning long term dependencies.

Overall, it was understood how chaos and stability of the dynamical system represented by an RNN are factors that affect the training, efficiency and long-range memorization ability. As a future direction, the model such as AS CF-RNN needs to improved based on stronger generalizations such as Lipschitz RNNs [9].

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

## A  Mathematical description of the networks

The mathematical equations corresponding to the computations involved in many architectures discussed in the main article are summarized in Table A.1. For each model specified in each row, the symbols for the different gates and states are listed in the last column. The hidden-to-hidden parameters are represented by $W$ matrices, the input-to-hidden parameters by $U$ matrices and the bias vectors by $b$.

In each case, $\odot$ denotes the element-wise (Hadamard) product, $\sigma$ is the sigmoid non-linearity and $tanh$ stands for the hyperbolic tangent non-linearity.

Table A.1: Specification of full equations of all the RNNs considered in the study

| Network | Equations | Variables |
|---|---|---|
| Long Short Term Memory (LSTM) [12] | $i_t = \sigma(W_i h_{t-1} + U_i x_t + b_i)$ 
 $f_t = \sigma(W_f h_{t-1} + U_f x_t + b_f)$ 
 $o_t = \sigma(W_o h_{t-1} + U_o x_t + b_o)$ 
 $g_t = tanh(W_g h_{t-1} + U_g x_t + b_g)$ 

 $c_t = \sigma(f_t \odot c_{t-1} + i_t \odot g_t)$ 
 $h_t = o_t \odot \sigma_h(c_t)$ | Gates: 
 $i_t, f_t, o_t$ 

 Cell input: 
 $g_t$ 

 States: 
 $h_t, c_t$ |
| Gated Recurrent Unit (GRU) [8] | $z_t = \sigma(W_z h_{t-1} + U_z x_t + b_z)$ 
 $r_t = \sigma(W_r h_{t-1} + U_r x_t + b_r)$ 

 $h_t = (1 - z_t) \odot tanh(U_h x_t + W_h(r_t \odot h_{t-1} + b_h))$ 
 $+ z_t \odot h_{t-1}$ | Gates: 
 $z_t, r_t$ 

 State: 
 $h_t$ |
| Chaos-Free Recurrent Neural Network (CF-RNN)[15] | $\theta_t = \sigma(W_\theta h_{t-1} + U_\theta x_t + b_\theta)$ 
 $\eta_t = \sigma(W_\eta h_{t-1} + U_\eta x_t + b_\eta)$ 

 $h_t = \theta_t \odot tanh(h_{t-1}) + \eta_t \odot tanh(U_h x_t)$ | Gates: 
 $\theta_t, \eta_t$ 

 State: 
 $h_t$ |
| Antisymmetric RNN (ASRNN) [6] | $h_t = h_{t-1} + \epsilon\, tanh((W_h - W_h{}^T - \gamma I)h_{t-1}$ 
 $+ V_h x_t + b_h)$ | State: 
 $h_t$ |
| Antisymmetric RNN - Gated (ASRNN-G) [6] | $z_t = \sigma((W_h - W_h{}^T - \gamma I)h_{t-1} + U_z x_t + b_z)$ 

 $h_t = h_{t-1} + \epsilon z_t \odot tanh((W_h - W_h{}^T - \gamma I)h_{t-1}$ 
 $+ U_h x_t + b_h)$ | Gate: 
 $z_t$ 

 State: 
 $h_t$ |
| Antisymmetric Chaos-Free RNN - Dual Gated (AS CF-RNN-DG) [proposed] | $\theta_t = \sigma((W_h - W_h{}^T - \gamma I)h_{t-1} + U_\theta x_t + b_\theta)$ 
 $\eta_t = \sigma((W_h - W_h{}^T - \gamma I)h_{t-1} + U_\eta x_t + b_\eta)$ 

 $h_t = h_{t-1} + \epsilon\, \theta_t \odot tanh((W_h - W_h{}^T - \gamma I)h_{t-1})$ 
 $+ \epsilon\, \eta_t \odot tanh(U_h x_t)$ | Gate: 
 $\theta_t, \eta_t$ 

 State: 
 $h_t$ |

# B   Jacobian computation for function $f$ in Section. 3

From eq. 3,

$$J(t) = \frac{\partial f}{\partial h_{t-1}} = \frac{\partial}{\partial h_{t-1}}\Big(\theta_t \odot tanh(h_{t-1}) + \eta_t \odot tanh(U_h x_t)\Big)$$

$$J(t) = \frac{\partial f}{\partial h_{t-1}} = \frac{\partial}{\partial h_{t-1}}\Big(\theta_t \odot tanh(h_{t-1})\Big) + \frac{\partial}{\partial h_{t-1}}\Big(\eta_t \odot tanh(U_h x_t)\Big)$$

$$J(t) = \left(\frac{\partial \theta_t}{\partial h_{t-1}} \odot tanh(h_{t-1}) + \frac{\partial tanh(h_{t-1})}{\partial h_{t-1}} \odot \theta_t\right) + \left(\frac{\partial \eta_t}{\partial h_{t-1}} \odot tanh(U_h x_t) + \frac{\partial}{\partial h_{t-1}}tanh(U_h x_t) \odot \eta_t\right)$$

$$J(t) = \left(\frac{\partial \theta_t}{\partial h_{t-1}} \odot tanh(h_{t-1}) + diag(tanh'(h_{t-1})) \odot \theta_t\right) + \left(\frac{\partial \eta_t}{\partial h_{t-1}} \odot tanh(U_h x_t)\right)$$

$$\implies J(t) = \left(\frac{\partial \theta_t}{\partial h_{t-1}} \odot tanh(h_{t-1})\right) + \Big(diag(tanh'(h_{t-1}) \odot \theta_t)\Big) + \left(\frac{\partial \eta_t}{\partial h_{t-1}} \odot tanh(U_h x_t)\right) \tag{B.1}$$

From eq. 1,

$$\theta_t = \sigma(W_\theta h_{t-1} + U_\theta x_t + b_\theta)$$

$$\implies \frac{\partial \theta_t}{\partial h_{t-1}} = diag(\sigma'(W_\theta h_{t-1} + U_\theta x_t + b_\theta)).\frac{\partial \theta_t}{\partial h_{t-1}}(W_\theta h_{t-1} + U_\theta x_t + b_\theta) \quad \text{(chain rule)}$$

$$\implies \frac{\partial \theta_t}{\partial h_{t-1}} = diag(\sigma'(W_\theta h_{t-1} + U_\theta x_t + b_\theta)).W_\theta$$

$$\implies \frac{\partial \theta_t}{\partial h_{t-1}} = diag\Big(\sigma(W_\theta h_{t-1} + U_\theta x_t + b_\theta) \odot (1 - \sigma(W_\theta h_{t-1} + U_\theta x_t + b_\theta))\Big).W_\theta$$

Substituting the sigmoids that evaluate to $\theta_t$,

$$\frac{\partial \theta_t}{\partial h_{t-1}} = diag\Big(\theta_t \odot (1 - \theta_t)\Big).W_\theta \tag{B.2}$$

Similarly for the derivative of $\eta_t$,

$$\frac{\partial \eta_t}{\partial h_{t-1}} = diag\Big(\eta_t \odot (1 - \eta_t)\Big).W_\eta \tag{B.3}$$

Using eqs. B.2 and B.3 in eq.B.1,

$$\boxed{\begin{aligned} J(t) &= diag\Big(\theta_t \odot (1 - \theta_t) \odot tanh(h_{t-1})\Big).W_\theta + diag\Big(tanh'(h_{t-1}) \odot \theta_t\Big) \\ &\quad + diag\Big(\theta_t \odot (1 - \theta_t) \odot tanh(U_h x_t)\Big).W_\eta \end{aligned}} \tag{B.4}$$

# C   Stability criterion for AS-RNN and variants

Figure C.1 shows the region on the complex plane of eigen values that the forward Euler method is stable. The circle of stability is shaded in pink and it is unstable elsewhere.

Theoretically, eigen values of a system in the entire left plane region corresponding to $Re(\lambda) < 0$ makes it stable. However, while using numerical methods such as Euler methods, the stability is affected by the numerical stability.

For forward Euler method to be stable, the condition to be satisfied by the eigen values of the Jacobian is the following (Proposition 2 in [6], originally in [2]):

$$\max_{i=1,2,...,n} |1 + \epsilon \lambda_i(\mathbf{J}(t))| \leq 1 \tag{C.5}$$

Any method using this scheme should constrain all eigen values inside this circle.

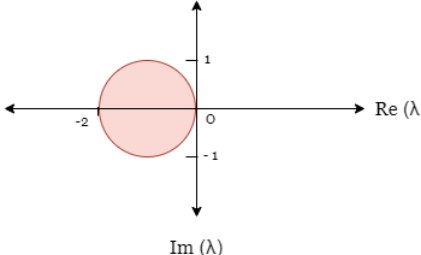

Figure C.1: Region of stability (indicated in pink) of an ODE with eigen values of the Jacobian $\lambda(\mathbf{J}(t))$. Outside the pink circle, the system is unstable under the forward Euler method.

# D   Hyperparameters used in experiments

## D.1   Analysis of Chaos

The parameter values for the networks used to plot the attractors, bifurcations and distance plot in Figure 1 are listed below for each network with hidden dimensions of 2. All other unspecified parameters were considered to be zeroes.

### D.1.1   LSTM

$$W_i = \begin{bmatrix} -1 & -4 \\ -3 & -2 \end{bmatrix} \quad W_o = \begin{bmatrix} 4 & 1 \\ -9 & -7 \end{bmatrix} \quad W_f = \begin{bmatrix} -2 & 6 \\ 0 & -6 \end{bmatrix} \quad W_g = \begin{bmatrix} -1 & -6 \\ 6 & -9 \end{bmatrix}$$

### D.1.2   GRU

$$W_z = \begin{bmatrix} 0 & 1 \\ -1 & 1 \end{bmatrix} \quad W_i = \begin{bmatrix} 0 & 1 \\ 1 & 0 \end{bmatrix} \quad U_z = \begin{bmatrix} 5 & -8 \\ 8 & 5 \end{bmatrix}$$

### D.1.3   CF-RNN

$$W_\theta = \begin{bmatrix} 0 & -1 \\ -1 & 0 \end{bmatrix} \quad W_\eta = \begin{bmatrix} 1 & 1 \\ 1 & 1 \end{bmatrix} \quad U_h = \begin{bmatrix} 1 & 0 \\ 0 & -1 \end{bmatrix}$$

## D.2   Copy Task

The most important hyperparameters used for the training of the models used in Section. 4.2 are tabulated below :

Table D.2: Architecture and training hyperparameters of the models used in the Copy Task

| | Architecture | | Training | |
|---|---|---|---|---|
| Model | $\gamma$ | $\|h\|$ | Optimizer | lr |
| AS CF-RNN | 0.005 | 128 | Adagrad | 0.01 |
| AS CF-RNN | 0.1 | 128 | Adagrad | 0.01 |
| AS-RNN (Gated) | 0.005 | 128 | Adam | 0.1 |
| AS-RNN (Gated) | 0.1 | 128 | Adam | 1 |
| AS-RNN | 0.005 | 128 | SGD | 0.001 |
| CF-RNN | - | 128 | Adam | 0.001 |
| LSTM | - | 128 | Adam | 0.001 |
| GRU | - | 128 | Adam | 0.005 |

The step size $\epsilon$ of the Aymmetric-matrix based models was chosen to be $0.01$.

# E    AS CF-RNN Experiments - Additional Results

Table E.3: Performance of the models in the permuted MNIST (pMNIST) Task

| | Architecture | | |
|---|---|---|---|
| Model | $\gamma$ | $\|h\|$ | Accuracy(%) |
| AS CF-RNN | 0.005 | 128 | 93.3 |
| AS-RNN (Gated) | 0.005 | 128 | 93.1 |
| AS-RNN | 0.005 | 128 | 95.8 |
| CF-RNN | - | 128 | 90.4 |
| LSTM | - | 128 | 92.6 |

