# OpenReview forum: "Recurrent Neural Networks as Dynamical Systems: Chaos and Order"
_umontreal.ca/Université_de_Montréal/Winter2021/MAT6215_

### Official Review · ~Antoine_Dion-St-Germain1 · 2021-05-02
**An improved Chaos-Free RNN with clear experimental results on sequential tasks**

**Rating:** 8
**Confidence:** 1

**Review:**

**Summary:**

*(1) Clarity :

The context is explained clearly. Every considered RNN is described in table A.1, which is helpful. The presentation of experimental results highlights the relevant conclusions.

*(2) Technical soundness :

The article has an adequate technical level and topic. More than one theoretical result are used to construct the proposed architecture. Each step of the reasoning is either explained or given with a source.

*(3) Impact :

It seems that the proposed architecture brings significant improvements to the other ones presented by the author.

*(4) Proposition :

The sequential task has been tested up to T = 800, which is the first value of T for which the accuracy drops bellow 100% for AS CF-RNN. It could be interesting to study the speed at which the accuracy drops as T grows compared to other RNNs.

**Pros :** Rich description of related works. Original results.

**Cons :** I don't think I understand the concept of long-range memorization ability.

**Section by Section Review:**

The article starts by explaining how chaos is an obstacle for RNNs and by precisely enumerating its content. It then shows how RNNs are related to dynamical systems, which is relevant for readers that are not familiar with the topic. The concept of Chaos-Free RNN (CF-RNN) is introduced as a way to stabilize the behaviours of a chaotic RNN. A few related results of other authors are presented, which helps illustrate the current state of the discipline. Table A.1 of appendix A contains a straightforward description of the considered RNNs.

In the following section, the author presents their proposed architecture in discrete time(eq. (3)), followed by continuous reformulation (eq. (4)). They then study the stability of the solutions by looking at the eigenvalues of the jacobian. All computation steps for the jacobian are shown in appendix B. The author explains why the real parts of the eigenvalues are all 0 using properties of antisymmetric matrices. They assume a diagonal matrix with no 0 on its diagonal. I am not familiar enough with the topic to evaluate if this assumption is legitimate in this context. A modified Euler method with a diffusion constant is used to solve the ODE. The use of a diffusion constant has been justified.

Section 4.1 shows experimental results that illustrates chaos in different RNNs. Figure 1 intuitively conveys that CF-RNN is the least chaotic tested RNN. It would have been interesting to see the result from the proposed AS CF-RNN next to those results. Section 4.2 challenges the AS CF-RNN with a sequential task. Table 1 shows clearly that it achieves high accuracy  and low cross-entropy error compared to other RNNs. Figure 2 shows the distribution of eigenvalues for the sequential tasks. The results are interpreted and a conclusion is made regarding the ability of AS CF-RNN just after.

The article ends a clear summary of what has been accomplished followed by a proposition for future work.

---

### Official Review · ~David_Yu-Tung_Hui1 · 2021-05-02
**Insightful dynamical systems analysis of chaotic RNN dynamics**

**Rating:** 8
**Confidence:** 4

**Review:**

Motivation

The relation between RNNs and dynamical systems is well motivated, with poor performance of baseline models in Table 1 compelling as well.  However, given the existence of CF and AS-RNNs, I thought the author could have motivated the need for their combination more.  I was uncertain over claims in the Background and Introduction setting that RNNs were remembering inputs and trying to achieve certain dynamics over input perturbations, as it seems that we would like an RNN to respond to inputs when it generalises from training to test data.

Soundness

I appreciated the concision of the author in the derivation, but it was hard to see why each step was justified until reading the whole section.  Although I was able to see how the author had combined AS-RNN and CF-RNN,I was uncertain why the latter was chosen (over the former) as the starting point.  It also seems that the use of a continuous ODE (Equation 4) was well justified in the results, and it would be nice to highlight this as it was introduced to put to rest any doubts the readers may have over why we were doing this over a discrete map (as RNNs work in discrete time).

Experiments

Results were impressive and clearly presented.  The graphs showed excellently that CF-RNN exhibited less chaos through learning a point attractor.  It would be nice to clarify that as a sub-variant, AS CF-RNN would also yield the same results.  Results in Table 1 were also very impressive and it would be good to see statistical t-tests to reinforce this by showing that AS CF-RNN is statistically significantly better than the baselines.  However, it was unclear whether Table 1 showed figures for the training or test set.  It seems like we did not see much evidence of exploding gradients, despite this mentioned in the abstract as an issue.

Writing

The work was well-structured overall and clearly written.  There were sections which felt undermotivated as I was reading them, and I felt that brief overviews at the beginning of each section could have alleviated this.  I felt that section 3 could benefit from closer links between text and math, for instance, writing the symbol names of the distinct gates for CF-RNN after they are first mentioned.  Unfortunately, the report is over four pages.

---

### Official Review · ~Laura_Estefany_Suarez1 · 2021-05-02
**Review on Recurrent Neural Networks as Dynamical Systems: Chaos and Order**

**Rating:** 8
**Confidence:** 4

**Review:**

**General comments:**

In this paper the author explores how stability criteria from dynamical systems theory can be used for the design of recurrent neural networks (RNNs). In the first part the author reproduces previous results that compare the dynamics of LSTMs and GRUs, which exhibit chaotic behaviour, against free-chaos RNNs, which have been designed to exclusively display stable dynamics. In the second part the author proposes a novel RNN architecture that combines the design principles that guarantee local stability in (single-gated) antisymmetric RNNs (AS-RNN) to improve (double-gated) free-chaos RNNs (FC-RNN). Results show that the new antisymmetric, free-chaos RNN (AS FC- RNN) architecture performs better against other architectures in a long-term memory task.

Conceptually speaking, the quality of the work is really good as it represents a successful case of how dynamical systems theory can be used to improve the design of RNNs. Accordingly, results from this research have a big potential for the fields of AI and machine-learning. Overall, the paper is well structured and clear, however, there are a few typos and redaction/style flaws that sometimes interfere with the flow of the story - working on those will significantly improve the quality and readability of the paper. The depth and breadth of the **Introduction** and **Background and Related Work** sections are sufficient to introduce the reader to the topic of RNNs, and how chaos affect their performance. Previous works on how to deal with chaos in RNNs described by the author also help to discern what are the specific contributions of this work, that is, how the new proposed architecture differs from previous ones. There are a few inconsistencies in the **Proposed Architecture** section related to the math derivations presented there (see specific comments below). In the **Results** section the author reproduces previous results on how chaos can be effectively suppressed in RNNs, and shows how the new proposed architecture can improve the ability of these networks to memorize long-term dependencies.


**Specific comments:**

- The new architecture proposed by the author uses two strategies to control the dynamics of the network: i) two gates: *input* and *forget*, and ii) constraining the parametric weight matrix to be antisymmetric. About the latter strategy, it is unclear whether the antisymmetric structure of the weight matrix is imposed before or after learning. If the former is true (i.e., the antisymmetric structure is imposed before learning), how does learning affect this structure? Is it preserved throughout training? To help clarify this point, it would be useful to compare the dynamics before and after learning.

- Related to the previous point, it is unclear what the dynamics of the new proposed architecture are, and how they compare to the dynamics of the two architectures it was initially inspired on, i.e., the AS-RNN and the FC-RNN. A figure similar to Figure 1 in which AS-FC-RNN dynamics are compared against AS-RNN and FC-RNN would help the reader to further understand the effects of the differences between architectures on the dynamics of the network. Are all these architectures equally effective in suppressing chaos?

- I believe there are some elements missing in the continuous-time formulation of the hidden state of the network $h_{t}$ at the end of page 2. If the continuous-time formulation is derived from the forward Euler method , i.e.:

$y_{t}$ =  $y_{t-1}$ + $\frac{dy}{dt}$*$dt$,

then the form of the derivative should look something like:

 $\frac{dy}{dt}$ = $\frac{y_{t} - y_{t-1}}{dt}$.

- The author presents some additional interesting results in the **Supplementary material**, but they are not mentioned at all in the main text.

**Future directions**:
- It would be interesting to perform the same type of analysis presented in this work using a task that requires a different type of computational property. For instance, a task that requires the recognition/classification of multiple temporal patterns. Would still suppressing chaos in this task contribute to improve performance? Are there tasks that are better supported by less stable dynamics?

---

### Official Review · ~Ryan_D'Orazio1 · 2021-05-03
**Interesting Combination of Chaos and Stability**

**Rating:** 8
**Confidence:** 2

**Review:**

**Overview**
RNNs are an important class of models used for data with sequential structure. Although these models have been always viewed as a dynamical system, recently connections with chaos, stability, and the performance of RNNs have been established and studied. In particular, the presence of
chaos and stability are seen as an inhibitor to learning. This work derives a novel architecture
combining previous ideas to avoid chaos and improve stability in RNNs. The resulting model grounded theoretical principles seems to be better than the sum of its parts, the original ideas regarding chaos and stability.

**Clarity**
Overall the report is well structured. Minor errors and typos aside the report reads well with clear motivation and  relevant references. However, it was  difficult to follow the equations
because I am not familiar with the notation and the RNN literature. I see there is some explanation in appendix, I would suggest to include the beginning of appendix A in section 3, and maybe add a little more. Though I also understand that 4 pages makes it difficult to include such details.

**Technical Details**
The technical details seem reasonable, and non-trivial. The approach is well motivated and interesting. Furthermore, the experiments and figures  paint a nice picture of existing work and the performance of the developed model  AS CF-RNN.

**Impact**
I believe this work can be impactful given the presented
results.

**Future Work**
I agree with the author, future experiments and investigation regarding generalization should be considered.

---

### Official Review · ~Simone_Totaro2 · 2021-05-03
**Recurrent Neural Networks as Dynamical Systems: Chaos and Order**

**Rating:** 8
**Confidence:** 3

**Review:**

### Summary
The authors argue about the importance of the RNN architecture from a dynamical system point of view.  The inductive bias introduced via a gating scheme can be analyzed via the "fixed input" dynamics. This technique, known in the literature, sheds light on the chaotic dynamics in architecture such as GRU and LSTM. Throughout the paper, the authors argue about the importance of two properties, namely chaos-free and local stability via the analysis of the continuous-time dynamical system induced by the RNN dynamics. Finally derives a novel architecture that borrows the best of both worlds.
Experimental results on the copy task, over long-horizon, seems to indicate that the new architecture can memorize better than the proposed benchmarks
## Novelty
The proposed architecture unifies stability from the AntiSymmetric and chaos-free dynamics from CF-RNN. The argument is to fight the vanishing gradient problem with the boundness of AntiSymmetric and global stability via the introduction of chaos-free gating.
The derivation of the architecture seems novel, but it is unclear what is the benefit of the union of these two ideas.
## Clarity
The paper is well written, slightly longer than 4 pages. In section 4.1 it seems that no training happens but the authors refer as if some learning should happen. A clear description of this experimental setting, as well as a comparison with AntiSymmetricRNN, would help understand the differences between the benchmark model and the proposed one.

### Questions:
The optimizer might hide some properties of the system. I wonder why there is a high starting learning rate for AntiSymmetricRNN?